# Estimating the Effect of Motivational Interventions in Patients with Eating Disorders: A Systematic Review and Meta-Analysis

**DOI:** 10.3390/jpm12040577

**Published:** 2022-04-04

**Authors:** Egzona Fetahi, Anders Stjerne Søgaard, Magnus Sjögren

**Affiliations:** 1Eating Disorder Research Unit, Psychiatric Centre Ballerup, 2750 Ballerup, Denmark; xona_dk@hotmail.com (E.F.); anders.stjerne.soegaard@regionh.dk (A.S.S.); 2Institute for Clinical Science, Umeå University, 901 85 Umeå, Sweden

**Keywords:** motivation, eating disorders, motivational interviewing, motivational enhancement therapy, anorexia nervosa, binge eating disorder, bulimia nervosa, EDNOS, OSFED

## Abstract

Motivation to change behavior is seen as an important factor in achieving a better treatment effect in patients with eating disorders (ED). The aim of this systematic review was to assess whether motivational interviewing (MI) and motivational enhancement therapy (MET) might (1) increase motivation to change behavior and (2) improve eating disorder psychopathology (EDP) and body mass index (BMI) in patients with ED. To investigate this, a literature search was conducted on 9 March 2021 on four scientific databases: Cochrane, Embase (Ovid), MEDLINE (PubMed), and PsycInfo (EBSCO). A total of 2647 publications were identified and following a rigorous stepwise procedure to assess titles and abstracts and, thereafter, full texts of relevant publications, 13 studies were included in the data extraction and analyses. A few individual studies (*n* = 5) found a significant increase in motivation, two a decrease in ED symptoms (*n* = 2), while none found an effect on BMI. However, the meta-analysis of each outcome found effect sizes near zero, thereby confirming the results of previous narrative reviews that have described a lack of effect of MET/MI on motivation in ED. Since the individual studies differ substantially in design, and the outcomes were inconsistently assessed with regards to instruments and duration, the effect of MET/MI on motivation for behavioral change, ED psychopathology, and BMI is still unclear.

## 1. Introduction

Eating Disorders (ED) such as Anorexia Nervosa (AN), Bulimia Nervosa (BN), and Binge Eating Disorder (BED) are serious health threats often affecting adolescents and young adults [1]. The prevalence of eating disorders in women in Europe is estimated to be 1–4% AN, 1–2% BN, and 1–4% BED, while the prevalence in men is 0.3% [2]. Chronicity is high in AN and BN with 20–23% remaining chronically ill [3,4], and, additionally, ED (especially AN) is known to have high mortality rates [5,6,7]. Due to the lack of evidence-based and effective treatments, there is a high unmet medical need for effective treatments of ED [8].

Patients suffering from ED are often ambivalent in their motivation for treatment [9,10], something especially described in AN but present in all ED, due to the egosyntonic nature of these disorders [10,11]. ED patients tend to use their disorder to cope with difficult feelings and often describe that the disorder gives them a sense of control [9,12]. Consequently, patients with ED often present with difficulties in changing their behavior [9,10]. Therefore, attempts to enhance motivation for behavioral change in ED, similarly as in abuse disorders [13], has been considered a relevant approach [9], and several studies have found a connection between the level of readiness to change and improvements in ED symptoms [14,15,16,17].

A model applied to explain the dynamic fluctuations of different stages of motivation for change is the Transtheoretical Model (TTM), as proposed by James Prochaska. TTM describes five “stages of change”: 1. Precontemplation (no intention to act), 2. Contemplation (considering change but not ready yet), 3. Preparation (ready and preparing to change), 4. Action (making changes in behaviors), and 5. Maintenance (changes have been made and focus is on maintaining these) [18]. The process is not linear but relapses into old behaviors and stages of change occur. In addition, attempts of intervening with individuals in the early stages of change may explain the lack of treatment effects [18,19].

Different types of interventions directed at enhancing motivation for behavioral change have been developed, perhaps the most well-known being Motivational Interviewing (MI) and Motivational Enhancement Therapy (MET); therapeutic interventions originally developed by William Miller and Stephen Rollnick to be used to enhance motivation to change in patients with substance abuse, and later tried in other disorders [19,20,21]. The principle of both MI and MET is to attempt to identify traces of ambiguous thoughts that the individual may be open to challenge. These ambiguous thoughts are the targets of the motivational enhancement interventions [19], which thereby increase the individuals’ intrinsic motivation [20].

The primary aim of this systematic review was to investigate the effects of MI and MET on motivation for behavioral change, ED psychopathology (EDP), and body mass index (BMI) in ED in comparison to various control conditions receiving, e.g., no intervention or other types of therapy.

## 2. Materials and Methods

### 2.1. Eligibility Criteria

The protocol was developed following the Preferred Reporting Items for Systematic reviews and Meta-Analyses for Protocols (PRISMA) guidelines and registered on PROSPERO: CRD42018098645. The eligibility criteria follow the PICO (Population, Intervention, Comparison, and Outcome) framework, with the population consisting of patients diagnosed with an ED according to criteria from the Diagnostic and Statistical Manual of Mental Disorders (DSM-IV or DSM-5) [22,23] or the International Statistical Classification of Diseases and Related Health Problems (ICD-10) [24], the intervention received being MET/MI as either a stand-alone intervention or incorporated into other interventions, while the comparator group was either on a waiting list or received other forms of active therapy. Eligible outcomes were one or more of the following: (a) Motivation, (b) EDP, and/or (c) BMI, and the study design had to be a randomized controlled trial (RCT) with the publication in English. For an overview of inclusion and exclusion criteria, see Appendix A.

### 2.2. Search and Selection

Literature search strategies were designed for each of the selected databases: Cochrane, Embase (Ovid), MEDLINE (PubMed), and PsycInfo (EBSCO). All searches were conducted on the 9 March 2021. Search terms included but were not limited to: “Anorexia nervosa”, “bulimia nervosa”, “binge-eating disorder”, “feeding and eating disorder”, “eating disorder not otherwise specified”, “EDNOS”, “other specified feeding and eating disorders”, “OSFED”, “unspecified food and eating disorder”, “UFED” “motivational interviewing”, “MI”, “motivational enhancement therapy”, “MET”, “motivational intervention”, “motivational level”, “motivational change counselling”, and “behavior change counselling”. The full search strategy is available in the Appendix A. Identification and removal of duplicates was performed manually in EndNote. The remaining titles and abstracts were screened using Rayyan [25], while full text publications were reviewed in EndNote. The screening was conducted by the 1st and 2nd authors (EF and ASS). The senior scientist (MS) was consulted in case of disagreement. A search for grey literature was conducted on WHO’s International Clinical Trials Registry Platform (ICTRP), International Standard Randomized Controlled Trial Number (ISRCTN), and ClinicalTrials.gov (accessed on 9 March 2021), in the reference list of all included studies and through a general Google search in order to find additional eligible studies.

### 2.3. Data Extraction

Clinically relevant information was extracted from the included publications and covered the following types of data: Identifying information (title, authors, country, publication year), research questions, participants (number of patients, age and gender, drop-out), diagnosis (type of ED, ED duration), treatment (ED treatment, inpatient/outpatient status, recruitment), study design and intervention (type, duration), outcome assessment details (instruments used to assess motivation, ED symptomatology and/or BMI, number of assessments, assessment time points), and outcome details (changes in level of motivation, ED behavior/symptoms or BMI). The full data extraction table can be found in Appendix B, Table A1.

### 2.4. Quality and Bias Assessment

A quality assessment was carried out using The Cochrane Risk of Bias 2 Tool (RoB 2) as all included studies were RCTs. RoB 2 consists of five domains: 1. Randomization bias, 2. Deviations from intended interventions, 3. Missing outcome data, 4. Measure of outcome, and 5. Selection of reported results [26]. Each of these domains received a rating, and then a final grading of the risk of bias was given: Low risk, some concern, or high risk. Protocols were needed for the assessments and searches were conducted on multiple sites to locate these. This included registries for clinical trials (clinicaltrials.gov, accessed on 9 March 2021; isrctn.com, accessed on 9 March 2021; trialsearch.who.int, accessed on 9 March 2021) and The Royal Library in Denmark (soeg.kb.dk, accessed on 9 March 2021). Requests for protocols were also sent by e-mail to study authors. The methods and results sections were compared if protocols were not available.

### 2.5. Data Analysis

The meta-analysis was conducted in the Review Manager 5.4 software (RevMan). Forest plots were created using the effect sizes: Mean difference (MD) and standardized mean difference (SMD or Hedges’ g). Pooled effect sizes were estimated, but it was not possible to conduct a subgroup analysis as too few studies were available. The mean change scores or the post-intervention scores were used to find MD and SMD and were calculated by subtracting the pre-intervention (baseline) scores from the post-intervention or follow-up scores [27]. As the included studies assessed outcomes at different time points, assessments made at time points furthest from the baseline were used to calculate the change scores. A maximal time interval of 6 months from the baseline was chosen to ensure that assessments were performed at similar time points, as a few studies had longer study periods and additionally performed assessments at, e.g., 12 months [27].

The mean change scores and their standard deviations (SD) were not available in all the included articles, and therefore these had to be calculated. This was performed using Pearson’s correlation coefficient (r), which is the correlation between the pre- and post-intervention scores, and the SD for both the baseline and post-intervention scores [27]. In the studies that only contained the standard error of the mean (SEM), the SD was calculated using SEM and the sample size (N) [27]. The correlation coefficients were not available in any of the included studies and were therefore requested from the authors, who had already provided other missing data. Professor Tracey Wade [28] and Dr. Danielle E. MacDonald [29] provided the correlation coefficients for the correlation between pre- and post-intervention scores for the Eating Disorder Examination (EDE), Eating Disorder Examination Questionnaire (EDE-Q), and BMI for participants in the intervention and control group (see Table 1). An average r was found for each outcome and used to calculate the SD for the change scores [27]. The calculated average r values can be found in Appendix B, Table A2. The correlation coefficients were not available for the relationship between pre- and post-intervention motivation scores, and therefore only the post-intervention scores were used in the meta-analysis for this outcome [27,30,31].

A sensitivity analysis was performed on the account of the imputation of the correlation coefficients, and therefore a range of correlation coefficients was used to assess the robustness of the pooled effect sizes: 0.3, 0.5, and 0.8. A coefficient of 0.3–0.5 indicates a low correlation, 0.5–0.7 a moderate, and 0.7–0.9 a high [26,32]. The heterogeneity between studies was explored using I^2^ (range: 0%–100%), which was interpreted based on the following definition: Low = 25%, moderate = 50%, and high = 75% [33].

A mixed-effects meta-regression was chosen to account for heterogeneity amongst the studies and was performed in R using the “metafor” package [34]. The chosen study-level moderators before conducting the analysis were: mean age, BMI, sample size, and ED duration. Data on ED duration were not available in all studies, and it was therefore not possible to test the effect of this moderator. R^2^-values from the meta-regression were used to determine whether any moderators could explain the heterogeneity [35]. Significance levels were set at *p* = 0.05 for all analyses in the meta-analysis and meta-regression. In case of missing data, the authors were contacted by e-mail to request data for completeness.

## 3. Results

### 3.1. Study Selection

In Figure 1, an overview of the study selection process is shown in the form of a PRISMA flowchart [36]. Through the literature search, 2638 publications were identified. After the screening of abstracts and titles, the full texts of 43 publications were retrieved and reviewed. Out of the 43 publications, 11 met the inclusion criteria, while 2627 publications were excluded. Two studies [37,38] were found when reviewing a protocol for a study by Schmidt et al. [39]. A search on the author’s name was conducted on PubMed to investigate if any articles had been published after the publication of the protocol. This search identified nine studies, and the same process, as described above, was performed with a screening of the abstracts, titles, and full texts, after which two studies were included. In the end, a total of 13 articles out of 2647 publications were included after deduplication, assessment, and reference searches.

### 3.2. Bias Assessment

The risk of bias was moderate to high in the included studies where three studies had some concerns [29,37,38], ten studies had a high risk of bias [16,28,40,41,42,43,44,45,46,47], while none of the studies had a low risk of bias. Two protocols [37,40] and four registries [29,38,43,47] were available for the bias assessment, while the remaining articles were assessed using the methods and results sections. The percentage of studies that scored a certain risk of bias level (low, some concern, high) for each domain can be seen in Figure 2. The bias assessment for the individual domains and the overall bias for each article are available in Figure 3.

### 3.3. The Effect of Motivational Interventions

#### 3.3.1. Study Characteristics

There was a total of 1322 participants in the 13 included studies. In the publications, where gender, age, and ED duration data were available, there were 885 women and 31 men in total, the participants’ mean age ranged from 19 to 42.5 years and the mean ED duration ranged from 3.16 to 15.1 years. The studies included different types of ED, where five studies only enrolled either AN [28,40,45], BED [41], or BN [16] patients, two studies included three different ED types (AN, BN, and EDNOS) [43,44], and the remaining six studies included two types of ED in different combinations of the following: AN, BN, BED, and EDNOS [29,37,38,42,46,47]. None of the studies included patients with Other Specified Feeding and Eating Disorder (OSFED) or Unspecified Feeding and Eating Disorder (UFED). Most participants received ED treatment concurrently with the interventions and were already enrolled in inpatient [28,45] or outpatient programs [37,38,40,43,46], with one study including both inpatients and outpatients [16]. Others did not receive ED treatment prior to or during the study, as they were either on a waitlist (WL) for treatment [29,44] or had been recruited from the broader community (a college, a university, and a large Canadian city) [41,42,47].

The incorporation of MET and MI in the interventions varied between studies. MET and MI sessions were given in combination with different elements, e.g., treatment-as-usual (TAU) [28,44], self-help (SH) manuals/handbooks [41,42,47], or as a part of programs, i.e., Motivation-Enhancing Psychotherapy for Inpatients with Anorexia Nervosa (MANNA) [45], MANTRA Maudsley Model of Anorexia Nervosa Treatment for Adults (MANTRA) [37,38], Recovery Maudsley Model of Anorexia Nervosa Treatment for Adults (RecoveryMANTRA) [40], and Readiness and Motivation Therapy (RMT) [43]. Three studies [16,29,40] had three study arms; Katzman et al. [46] and Treasure et al. [16] had two groups receiving MET in combination with either group or individual Cognitive Behavioral Therapy (CBT), while MacDonald et al. [29] had one of three arms receiving MI in combination with TAU. RecoveryMANTRA used in Cardi et al. differed from the rest as the intervention group received text–chat sessions based on MI instead of face-to-face interventions. The comparator groups also received various types of interventions SH [41,42], SH and psychoeducation (PE) [47], TAU [28,40,44,45], CBT [16,29,46], and Specialist Supportive Clinical Management (SSCM) [37,38], or WL [43].

The intervention frequency ranged from a singular MI and MET session to long study periods with one study offering up to 36 sessions (e.g., MANTRA), while the length of the therapy sessions was 60–80 min. The interventions were performed by therapists with differing training and backgrounds. Some therapists had no prior knowledge about MET/MI and others were experienced therapists. Both passive methods of training, i.e., learning about MET/MI through books or training videos/audios, and active methods, i.e., role-playing, full day training camps, or repeated training sessions on study participants, were used with most studies using a combination of these methods. Assessments and/or supervision of the therapists were performed using audio or video recordings in some studies, while others assessed or supervised the therapists in person.

A wide variety of measurement tools were used to assess the outcomes (motivation, EDP, and BMI), and not all studies measured outcomes at the same time points. Post-intervention and follow-up assessment time points ranged from weeks to 12 months. Motivation to change was assessed using common and validated questionnaires in some of the included studies: University of Rhode Island Change Assessment Scale (URICA) [16,42,46,47], the short form of the URICA (URICA-S) [45], The Autonomous and Controlled Motivations for Treatment Questionnaire (ACTMQ) [40], Anorexia Nervosa Stages of Change Questionnaire (ANSOCQ) [28], and a semi-structured Readiness and Motivation Interview (RMI) [43]. Other studies used unvalidated instruments to assess motivation: The motivation to change scale (MTC), which consisted of three questions and 10-point Likert scales [44], six self-report questions [28], two visual analogue scales [40], and “change ratings” made up of three questions and a visual analog scale [41]. In the rating of ED symptoms most studies used validated questionnaires in the form of: The Eating Disorder Examination Questionnaire (EDE-Q) [29,37,40,42,45,47], the semi-structured interview Eating Disorder Examination (EDE) [28,29,37,38], the self-report questionnaire Eating Disorder Inventory-2 (EDI-2) [43], the self-report Pros and Cons of Eating Disorders Scale (PCED) [44], the Psychiatric Status Rating (PSR) [45], and the self-report Weight Efficacy Lifestyle (WEL) questionnaire [41,47]. However, unvalidated instruments were also used: An unnamed scale [16], a standardized semi-structured interview for suitability (elements from LIFE and EDE) [46], and the Timeline Follow-Back Interview [41]. BMI was measured, derived, or recorded either by self-report, the study team, the ED treatment clinicians or from patient records.

#### 3.3.2. Outcomes

##### Motivation to Change

Motivation was assessed in 10 studies [16,28,40,41,42,43,44,45,46,47], where eight of these [16,28,40,42,43,44,45,47] measured motivation at baseline and post-intervention. Cassin et al. [41] also assessed motivation at different post-intervention time points but did not assess motivation at baseline. Five out of these nine studies [40,41,42,43,47] saw a statistically significant improvement in some aspect of motivation to change in the intervention group compared to the control groups.

Three studies used a motivational intervention in combination with SH (MET/MI + SH) [41,42,47]. Vella-Zarb et al. [47] found a significant increase in readiness to change for MI + SH compared to controls (PE + SH) (t(23) = −4.11, *p* < 0.001, d = 0.61). Dunn et al. [42] found that contemplation and action scores significantly increased in the MET + SH compared to the control group, that only received SH (t(44) = −0.36, *p* < 0.01, d = 0.42). Cassin et al. [41] found that the MI + SH group had significantly higher scores in confidence to change at post-intervention when compared to the SH group (t(106) = 4,91, *p* < 0.01). In the study by Cardi et al. [40], RecoveryMANTRA significantly increased confidence in the ability to change at post-intervention compared to TAU (t(184) = −2,41, *p* = 0.02). However, this difference was not significant at follow-up. Geller et al. [43] found that the RMT group was significantly less ambivalent about change compared to waitlist controls at both follow-up points ($X^{2}=4.08$, *p* < 0.05 and $X^{2}=3.94$, *p* < 0.01). However, all these studies had a high risk of bias.

##### Eating Disorder Psychopathology

ED symptoms and behaviors were measured in all 13 studies, 12 studies measured EDP at both baseline and post-intervention [16,28,29,37,38,40,42,43,44,45,46,47]. Cassin et al. did not measure EDP at baseline but instead measured the change over several post-intervention time-points [41]. Out of 12 studies, 3 [29,41,42] found a significant improvement in ED symptoms at post-intervention when comparing groups. MacDonald et al. [29] found that patients receiving CBT-RR had significantly fewer total episodes of bingeing/vomiting/laxative use over the first four weeks of treatment (t(24.70) = 2.40, *p* = 0.02, d = 0.75), but this difference could not be seen at the end of treatment. Moreover, they significantly improved on the “*overvaluation of weight and shape*”-subscale in the EDE-Q (*p* = 0.008) compared to the group receiving MI. It should, however, be noted that MacDonald et al. had some concerns in the assessment of bias.

The MET/MI group performed better than the control group in the other two high risk studies by Cassin et al. [41] and Dunn et al. [42]. Cassin et al. found that significantly more participants in the AMI group were confident in their ability to resist overeating. Furthermore, the binge eating frequency was reduced significantly in the AMI group at all follow-up points (t(106) = 3.91, *p* < 0.001, d = 0.76; t(106) = 2.88, *p* = 0.005, d = 0.56; t(106) = 4.11, *p* < 0.001, d = 0.80), and also more patients in this group abstained from bingeing ($X^{2}$(1, N = 108) = 4.79, *p* = 0.03). Moreover, fewer AMI participants met the frequency criterion for BED from DSM-IV ($X^{2}$(1, N, = 108) = 11.82, *p* = 0.001). Dunn et al. found that MET + SH significantly increased the number of participants who abstained from bingeing after 4 months ($X^{2}$(1, N = 90) = 3.92, *p* < 0.05).

##### Body Mass Index

BMI was measured or recorded at baseline in all 13 studies, but only five studies measured BMI at pre- and post-intervention [28,37,38,40,45]. All five studies included patients with AN with none finding a significant difference between the intervention and control groups. The studies had a moderate [37,38] to high risk of bias [28,40,45].

### 3.4. Data Analysis

#### 3.4.1. Meta-Analysis

A meta-analysis was conducted on measurements from 651 participants using nine out of the 13 included studies [28,29,37,38,40,41,42,45,47]. For all outcomes the effect sizes were near zero and non-significant, showing that there was no difference between the groups for any of the outcomes. Of the included studies, three measured motivation using a comparable method at post-intervention and had an SMD = 0.03, which was non-significant (*p* = 0.82; 95% CI = (−0.25;0.31)). The level of heterogeneity between the studies was low (I^2^ = 0%; *p* = 0.63) (Figure 4). Mean change scores on EDP were compared using eight studies with an SMD = 0.02 (*p* = 0.85; 95% CI = (−0.20;0.24)), and the level of heterogeneity was moderate (I^2^ = 43%; *p* = 0.09) (Figure 5). In the sensitivity analysis for this outcome, the effect sizes (SMD) ranged from 0.01 to 0.02, and the results were all non-significant (*p* = 0.80–0.94) (Appendix A). The analysis for BMI included change in mean BMI from six studies, which showed an MD = −0.04 (*p* = 0.68, 95% CI = (−0.21;0.14)). The level of heterogeneity was once again low (I^2^ = 1%; *p* = 0.41) (Figure 6). In the sensitivity analysis, effect sizes (MD) ranged from −0.01 to 0.01 and were all non-significant (*p* = 0.95–0.97) (Appendix A).

#### 3.4.2. Meta-Regression

The studies measuring EDP were used to perform a meta-regression analysis, as several studies measured this outcome. Data on three variables are available: Mean age (Appendix A), BMI (Appendix A), and sample size (Appendix A). None of these variables could account for the variance between studies with all showing R^2^ = 0 and *p* > 0.05. The analysis data with bubble plots are available in Appendix A.

## 4. Discussion

The aim of this systematic review and meta-analysis was to investigate and quantify the effect of motivational interventions, i.e., MET and MI, on motivation for behavioral change, EDP, and BMI in patients with ED by using a comparator group with ED patients receiving a control intervention. A few of the included studies found that MET and MI, at an individual study level, elicited a significant positive change in either motivation or EDP; although, there were none in BMI. However, the meta-analysis showed no overall difference in the effect on any of the aforementioned outcome measures in the intervention groups compared to the control groups, with all the pooled effect sizes being near zero. This suggests that motivational interventions in the form of MET and MI are not better than other therapeutic interventions when used on ED patients.

These results are in line with earlier systematic reviews on this topic. In a review by Knowles et al. [48], the included studies did not seem to favor motivational interventions, while one by Macdonald et al. [49], which simultaneously looked at the effect on carers, found that there may be a place for motivational interventions, e.g., as an introductory intervention. These systematic reviews differ from the current by including a meta-analysis, thereby enabling a collated, parametric comparison of the included studies. The results of this meta-analysis strengthen the notion that the studies performed up till now using motivational interventions in ED do not support an effect on motivation for behavioral change, EDP, nor BMI when compared to other types of therapies.

### 4.1. The Effect of MET/MI on Motivation to Change

In the meta-analysis, the pooled effect for the effect of motivational interventions on motivation to change in patients with ED was near zero (SMD = 0.03; *p* = 0.82; 95% CI = (−0.25;0.31)), showing a lack of difference in the effect in the group receiving motivational interventions compared to control groups. On a study level, only five of the included [40,41,42,43,47] studies saw a significant change in motivation. However, different measurement tools were used in the respective studies, assessing different aspects of motivation, making it nearly impossible to derive a conclusion regarding the effect. Furthermore, the studies used motivational interventions that consisted of low-intensity interventions (few MET/MI sessions or a text-based intervention), and the control groups were either not receiving treatment (WL) or received low-intensity treatment, e.g., SH or PE. This could explain the observed significant difference found on a study-level between the groups in these particular studies, as earlier reviews have shown that MI or MET does cause a significant change in motivation to change when compared to low-intensity interventions [48,49].

The lack of effect in the meta-analysis could have been caused by several factors. Notably, the availability of data was an issue, with only three of the included studies having comparable motivation scores, and since baseline scores were not available for all studies, the post-intervention scores were used instead of the mean change from baseline. In addition, the heterogeneity between studies could have influenced the results. They differed in the types of ED enrolled, recruitment settings, outcome measures used to assess motivation to change, types of motivational intervention programs, and control groups used. Furthermore, the three studies in the meta-analysis all had a high risk of bias. Wade et al.’s randomization process was unclear, and both Cassin et al. and Wade et al. used unvalidated outcome measurement instruments, and, therefore, these domains were at a high risk of bias. Furthermore, all three studies only had one domain with a low risk of bias, which suggests that the study design could potentially have influenced the results in the individual studies.

### 4.2. The Effect of MET/MI on Eating Disorder Psychopathology

The meta-analysis found that the pooled effect size of the effect of motivational interventions on EDP was near zero (SMD = 0.02; *p* = 0.85; 95% CI = (−0.20;0.24)). At an individual study level, only two studies by Cassin et al. [41] and Dunn et al. [42] found a significant improvement of ED symptoms in the intervention group compared to the control group. Additionally, these two studies were the only studies that saw an improvement in multiple outcomes (motivation for behavioral change and EDP).

The meta-analysis for this outcome was robust, compared to the other analyses in this review, as a greater number of studies were included. Additionally, half of the included studies, i.e., the studies by MacDonald et al. [29], Schmidt et al. (2012) [38], Schmidt et al. (2015) [37], and Ziser et al. [45], used intensive motivational interventions consisting of multiple MET or MI sessions per week over longer periods of time (weeks to months), with longer follow-up periods (from weeks up to 1 year), and compared these to equally intensive interventions in the control groups. Furthermore, the studies all measured changes in EDP with EDE-Q and/or EDE, making a comparison of the effects on the outcomes easier. Although the risks of bias in the studies by Schmidt et al. (2012), Schmidt et al. (2015), and MacDonald et al. were of some concern, these studies had the most domains with low risk of bias, and they were the most detailed and thorough of all the studies included in this review. A potential explanation for the lack of effect, despite the inclusion of studies of higher quality, could be that the control groups were receiving therapeutic interventions equally as intensive as that of MET/MI, an explanation also proposed in earlier publications [48].

### 4.3. The Effect of MET/MI on Body Mass Index

The meta-analysis showed a pooled effect size near zero (MD = −0.04; *p* = 0.68; 95% CI = (−0.21;0.14)), and furthermore, none of the studies found a significant between-group difference in BMI on an individual study level. All included studies measured BMI, which is used as a measure of disease severity in AN and could explain why these particular studies measured BMI at both pre- and post-intervention, as they all enrolled AN patients. The only exception was the study by MacDonald et al. [29], where BN and PD patients were enrolled instead. Therefore, the result from this analysis is virtually limited to patients with AN and suggests that neither MI nor MET can be used to increase the BMI of AN patients. As suggested above, many of the included studies used intensive interventions in both active and comparator treatments, which may explain the lack of a superior effect of MET/MI to the comparator group.

### 4.4. Strengths and Limitations

The current systematic review has some strengths; i.e., the inclusion of only RCT, meaning that all studies had a control group, as well as the change in mean scores in the meta-analysis for two of the outcomes (EDP and BMI).

However, there are also some limitations such as that the meta-analysis meant applying an imputation of correlation coefficients. Although the sensitivity analysis found similar pooled effect sizes, regardless of the size of the correlation coefficient, the use of the original correlation coefficients from the respective studies would have led to the most accurate pooled effect sizes. Furthermore, not all studies could be included in the meta-analysis, and therefore it was not possible to calculate the change in mean scores from baseline to post-intervention for all the included studies and compare these to the change in mean from baseline to different follow-up time points. Consequently, only the scores furthest away in time from baseline could be included, where 6 months from baseline was chosen as the maximal time interval to ensure that assessment time points between studies were as close to each other as possible. Additionally, it was not possible to make a subgroup analysis due to the low number of eligible studies, and although most studies accounted for drop-out in their analyses, it was not possible to account for this for all studies, as some only provided data in the form of a completer analysis. Consequently, the low number of included studies, the high drop-out rates, and the small sample sizes may have led to an underpowering, which may have influenced the validity of the meta-analysis and the results in the individual studies.

Another limitation was the heterogeneity between studies with interventions differing in the type, application, and duration of the interventions. Furthermore, the outcomes were measured at differing frequencies and time points using different types of assessment instruments with some studies using subscale scores and others global scores, some using unvalidated or validated assessment instruments for patients with ED, and almost all studies using self-report measures in some capacity with a potential risk of measurements being imprecise or unreliable. Moreover, the generalizability of the results was limited as most studies used participants consisting solely or mainly of women. In addition, the risk of bias assessment showed that most studies had some degree of methodological issues as none of the studies scored a low risk of bias.

### 4.5. Future Studies

Future clinical trials on motivational interventions in ED would benefit from including more patients to improve the sensitivity to detect the true effects of motivational interventions on motivation for behavioral change, ED symptoms, and BMI (in AN). Furthermore, there is a need for studies that use similar MET and MI interventions, preferably as stand-alone interventions, as this would improve the ability to discern an effect of these interventions. Moreover, there is room for improving future studies by ensuring that validated outcome instruments are used to enable proper comparison between studies, and it is additionally warranted that studies assess outcomes at multiple time points and make use of longer follow-up periods.

## 5. Conclusions

The results from this systematic review and meta-analysis suggest that there is no effect of motivational interventions on motivation for behavioral change, eating disorder psychopathology, and body mass index compared to control interventions. However, the large degree of heterogeneity between studies and the low number of studies with available data to use for comparisons should be considered when interpreting the results. Future studies would benefit from having larger sample sizes, motivational interventions as a stand-alone intervention, similar outcome instruments, longer follow-up periods, and from measuring the outcomes at all time points.

## Figures and Tables

**Figure 1 jpm-12-00577-f001:**
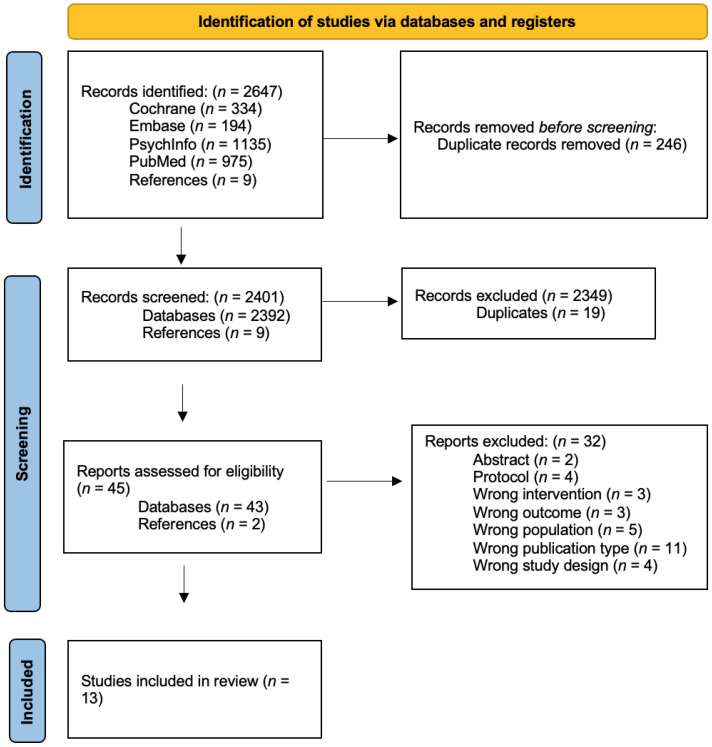
Preferred Reporting Items for Systematic Reviews and Meta-Analyses (PRISMA) flowchart showing the steps of study inclusion [36].

**Figure 2 jpm-12-00577-f002:**
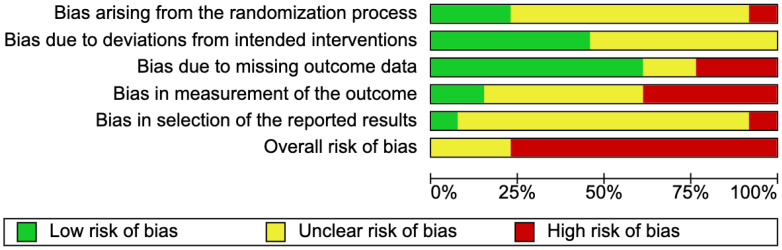
Graph showing the percentage of risk of bias scores for each domain of all included studies.

**Figure 3 jpm-12-00577-f003:**
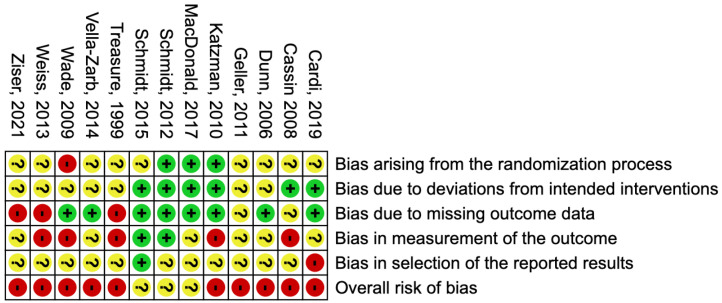
Summary table showing the risk of bias assessments for each domain and overall, for all included studies [16,28,29,37,38,40,41,42,43,44,45,46,47].

**Figure 4 jpm-12-00577-f004:**
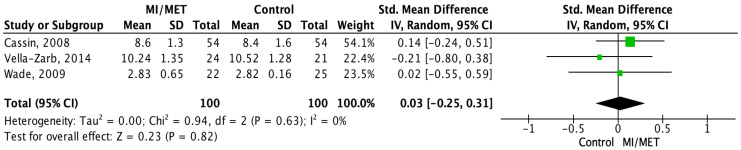
Forest plot displaying the pooled effect of motivational enhancement therapy/motivational interviewing (MET/MI) on motivation using the standardized mean differ-ence (SMD) [28,41,47]. Mean motivation scores at post-intervention (mean), Standard Deviation (SD).

**Figure 5 jpm-12-00577-f005:**
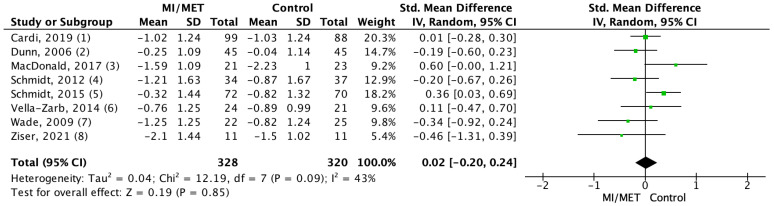
Forest plot displaying the pooled effect of MET/MI on eating disorder psychopathology (EDP) using SMD [28,29,37,38,40,42,45,47]. Mean change in EDP scores from baseline to follow-up (mean).

**Figure 6 jpm-12-00577-f006:**
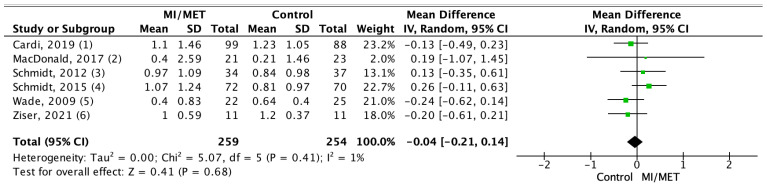
Forest plot displaying the pooled effect using the mean difference (MD) of MET/MI on BMI [28,29,37,38,40,45]. Mean change in BMI from baseline to follow-up (mean).

**Table 1 jpm-12-00577-t001:** The correlation coefficients (Pearson’s r) provided by MacDonald et al. [29] and Wade et al. [28].

Article	BMI_Control_	BMI_Intervention_	EDP_Control_	EDP_Intervention_
Pearson’s r (MacDonald et al.)	0.997	0.988	0.28 (EDE-Q)	0.24 (EDE-Q)
Pearson’s r (Wade et al.)	0.75	0.42	0.77 (EDE)	0.72 (EDE)

Abbreviations: Eating Disorder Examination (EDE), Eating Disorder Examination Questionnaire (EDE-Q).

## Data Availability

Not applicable.

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
