# Peer review of "Estimating the Effect of Motivational Interventions in Patients with Eating Disorders: A Systematic Review and Meta-Analysis"

_jpm, 2022, doi:10.3390/jpm12040577_

Round 1

Reviewer 1 Report

A very interesting meta-analysis, well presented and described. In general, the approach and objectives are well understood, and the results respond perfectly to them.

My comments are minor:

  • Line 67, they could include some example of what they consider control condition.
  • Line 75, the diagnosis with DSM-IV is applicable to the DSM-5 criteria?
  • Table 1 needs a title and indicate in the footer what EDE-Q and EDE mean.
  • Revise section 3.1. to match the sample sizes in Figure 1. Were there any temporal criteria in the search for articles? (e.g. from 2000 to the present or similar?)
  • The results are greatly altered by excluding the Treasure, 1999 article? it is the oldest and most biased.
  • Line 211 and 212, defines OSFED and UFED.
  • Bold in the text is not necessary.
  • Line 289, article 42 is not referenced in line 288.
  • Line 311, there is a "(".

Author Response

Dear Reviewer 1

Thank you for the feedback on our article! In the following, you will find our responses to the suggested changes.

Comment 1: Line 67, they could include some example of what they consider control condition.

Response 1: Thank you for this suggestion. We agree and have updated the manuscript which can be seen in lines 67-68.

Comment 2: Line 75, the diagnosis with DSM-IV is applicable to the DSM-5 criteria?

Response 2: Thank you for this comment. In our inclusion criteria, we accepted publications using either DSM-IV or DSM-5, and therefore, we have changed the sentence in line 76 to describe this.

Comment 3: Table 1 needs a title and indicate in the footer what EDE-Q and EDE mean.

Response 3: Thank you for pointing this out. Table 1 was accidentally deleted before uploading the Word file for submission. We have inserted Table 1 again with a title and explanations of the abbreviations have been added in the footer. These changes can be seen in lines 153-156.

Comment 4: Revise section 3.1. to match the sample sizes in Figure 1. Were there any temporal criteria in the search for articles? (e.g. from 2000 to the present or similar?)

Response 4: This comment is greatly appreciated. As suggested by the first half of the comment we have revised section 3.1 and Figure 1 to aid understanding of the search and selection process. Regarding the second half of the comment, we had no temporal criteria in our eligibility criteria. All eligibility criteria can be seen in Table S1 in the supplementary materials.

Comment 5: The results are greatly altered by excluding the Treasure, 1999 article? it is the oldest and most biased.

Response 5: Thank you for this comment. We have included all eligible publications in the meta-analysis as is customary. In the abstract and the discussion, we have commented on the heterogeneity of the included studies which may influence the results. Furthermore, we have also suggested the need for further, more homogeneously designed studies.

Comment 6: Line 211 and 212, defines OSFED and UFED.

Response 6: Thank you for pointing this out. We have now defined both abbreviations in the text in lines 214-215 and under “Abbreviations”.

Comment 7: Bold in the text is not necessary.

Response 7: Thank you for this reminder. We have removed bold words in the text throughout the manuscript except for the headings.

Comment 8: Line 289, article 42 is not referenced in line 288.

Response 8: Thank you for pointing this out. Article 42 is not referenced here, as this publication did not assess the outcome at baseline unlike the rest of the publications referenced in line 292. We have revised the sentence and to not cause further confusion, we have added an explanation in lines 292-294.

Comment 9: Line 311, there is a "(".

Response 9: Thank you for pointing out this error, the "(" has been deleted in line 316.

Reviewer 2 Report

This manuscript is written well, please do spell-check.

Author Response

Dear Reviewer 2

Thank you for taking the time to review our manuscript!

Comment: This manuscript is written well, please do spell-check.

Response 1: Thank you for your kind comment. We have done spell-check and the changes to the manuscript have been tracked in Word.